# ctDNA as an Objective Marker for Postoperative Residual Disease in Primary Advanced High-Grade Serous Ovarian Cancer

**DOI:** 10.3390/cancers17050786

**Published:** 2025-02-25

**Authors:** Valentina Glueck, Christoph Grimm, Magdalena Postl, Christian Brueffer, Nuria Segui, Miguel Alcaide, Lucia Oton, Yilun Chen, Lao H. Saal, Gerda Hofstetter, Stephan Polterauer, Leonhard Muellauer

**Affiliations:** 1Gynecologic Cancer Unit, Division of General Gynecology and Gynecologic Oncology, Department of Obstetrics and Gynecology, Comprehensive Cancer Center, Medical University of Vienna, 1090 Vienna, Austria; christoph.grimm@meduniwien.ac.at (C.G.); magdalena.postl@meduniwien.ac.at (M.P.); stephan.polterauer@meduniwien.ac.at (S.P.); 2Department of Obstetrics and Gynecology, Klinikum Starnberg, 82319 Starnberg, Germany; 3SAGA Diagnostics AB, 223 81 Lund, Sweden; christian.brueffer@med.lu.se (C.B.); nuria.segui@sagadiagnostics.com (N.S.); miguel.alcaide@sagadiagnostics.com (M.A.); lucia.oton@sagadiagnostics.com (L.O.); yilun.chen@sagadiagnostics.com (Y.C.); lao.saal@med.lu.se (L.H.S.); 4Division of Oncology, Lund University Cancer Center, Skåne University Hospital Comprehensive Cancer Center, Lund University, 221 00 Lund, Sweden; 5Department of Pathology, Medical University of Vienna, 1090 Vienna, Austria; gerda.hofstetter@meduniwien.ac.at (G.H.); leonhard.muellauer@meduniwien.ac.at (L.M.)

**Keywords:** ovarian cancer, circulating tumor DNA (ctDNA), minimal residual disease (MRD), objectification of postoperative tumor residual disease

## Abstract

The standard of care for assessing postoperative residual disease in advanced epithelial ovarian cancer is the subjective intraoperative evaluation of the surgeon. The aim of our prospective study was to demonstrate the feasibility of perioperative tumor-informed circulating tumor DNA (ctDNA) levels as a marker for postoperative residual disease (RD). This is of utmost importance as complete resection is currently a key factor for the patient’s further prognosis. Our personalized ctDNA approach provided extraordinarily high detection rates pre- and postoperatively (96% and 81%, respectively). ctDNA levels decreased significantly after complete tumor resection but to lesser extent in patients with higher tumor stage (≥FIGO3C). ctDNA levels can reflect the postoperative tumor residuals and can provide a further prognostic indicator.

## 1. Introduction

Ovarian cancer (OC) is the most lethal tumor in gynecological malignancies [1,2,3,4]. The most common subtype is epithelial ovarian cancer (EOC) [5]. The cornerstone of primary treatment is surgery with complete tumor resection, followed by adjuvant chemotherapy and subsequent maintenance therapy [6,7,8,9]. Evaluation of postoperative residual disease (RD) and response to chemotherapy seem to be the most important prognostic factors for these patients [10].

The prognostic value of complete tumor resection has been confirmed in several trials with significant impact on outcome [11]. The standard for classifying the amount of postoperative RD is the surgeon’s evaluation at the end of surgery. However, it cannot be ruled out that on the one hand the evaluation remains subjective and on the other hand a more precise mode of assessment of minimal residual disease (MRD) is required. All currently clinically available tests, such as postoperative CA-125 serum levels or imaging, are unreliable in reflecting postoperative residual disease. This applies particularly for MRD [12,13].

Most recently, attention has been drawn to circulating tumor DNA (ctDNA) for the evaluation of MRD in a variety of malignancies [14,15,16,17,18,19]. In general, ctDNA levels are more elevated in patients with advanced stages of breast, colorectal, pancreatic, and esophageal cancer than in early-stage tumors [20,21,22] and are generally well correlated to the tumor load. The capabilities of ctDNA were recently acknowledged in a statement by the U.S. National Cancer Institute Task Force, as the presence of ctDNA is strongly associated with high risk of disease recurrence in colorectal cancer patients [23]. Evidence supporting the concept that ctDNA is a robust marker for postoperative MRD is rapidly growing [24]. Recent studies underline the potential of ctDNA as a diagnostic tool for EOC [25,26,27,28,29,30,31,32,33,34] and demonstrate clinical superiority compared to the conventional biomarker CA-125 [28,35,36,37,38].

Correlation of ctDNA dynamics with response to adjuvant chemotherapy was demonstrated and may predict progression or response earlier than CA-125 [39] or computed tomography (CT) [40]. CtDNA analysis demonstrates high correlation with mutations and epigenetic changes seen in tumor biopsies and could be used for treatment response monitoring and detection for MRD. Of note, the majority of these approaches used tumor-naïve ctDNA approaches and therefore showed limited detection rates of ctDNA in EOC [41].

Despite the growing literature for ctDNA and evaluation of MRD, there is currently limited data [42] describing the value of a precise tumor-informed ctDNA approach as a measurement for postoperative RD and MRD in EOC. Thus, we investigated the feasibility and accuracy of a tumor-informed personalized ctDNA approach utilizing monitoring of somatic structural rearrangements and single nucleotide variants (SNVs) in advanced EOC patients for the measurement of postoperative RD, comparing it with the current gold standard, i.e., the surgeon’s evaluation at the end of debulking surgery.

## 2. Materials and Methods

Patients with primary advanced EOC who underwent surgery from July 2021 to July 2022 at the division of Gynecologic Oncology at the Medical University of Vienna (MUVI), Austria were included in this prospective study. Informed consent was provided by all participating patients at inclusion. Approval of the institutional ethics committee was obtained before initiation of the study (EK-No: 1502/2020). This study was conducted according to ethical standards and guidelines within the Declaration of Helsinki and according to good scientific practice guidelines of the Medical University of Vienna. All patients received therapy according to current international guidelines. Postoperative RD was independently assessed by two experienced gynecological surgeons.

Tumor tissue was assessed intraoperatively during primary surgery and collected as formalin-fixed, paraffin-embedded (FFPE), or fresh frozen material. Blood plasma sampling for ctDNA analysis was performed preoperatively and on the second (d2) and tenth day (d10) postoperatively. Two “Cell-Free DNA Collection Tubes” from Roche Diagnostics (Basel, Switzerland), each with a tube volume of 8.5 mL, were taken at each blood sampling time. Blood samples/biomaterial were processed and stored according to standard operating procedures by the Medical University of Vienna Biobank in an ISO 9001:2015-certified environment [43].

In the present study, 66 patients with OC were screened. Of these, 39 (59.1%) had to be excluded due to histology other than EOC (*n* = 31) or lower tumor stage (*n* = 3). In four patients no blood samples were taken postoperatively due to early discharges from the hospital. One patient had undergone extensive prior surgery at another clinic, so there was hardly any tumor tissue present at the time of the operation in our clinic. Thus, samples from 27 patients were available for final analysis to compare ctDNA levels before and after surgery (see CONSORT diagram/Figure 1).

We used a new and unique test protocol for detection of ctDNA in patients with advanced EOC. The ctDNA assessment was based on a tumor-informed ctDNA approach (SAGA Diagnostics AB, Lund, Sweden). Low-coverage whole genome sequencing (WGS) (3–23× mean sequencing depth) was used to identify structural variants (SVs), single nucleotide variants (SNVs), and indels in tumor tissue in order to develop personalized digital polymerase chain reaction (dPCR) assays.

### 2.1. DNA Extraction

For 15 of the 27 patients, DNA was extracted from fresh frozen tumor tissue obtained from one or several tumor locations per patient using the AllPrep DNA/RNA Mini Kit (Qiagen, Hilden, Germany, Cat. No. 80204) and following the manufacturer’s instructions for DNA-only extraction. For the remaining 12 patients, FFPE tumor tissue was processed for DNA extraction using the truXTRAC FFPE total NA Ultra Kit (Covaris, Woburn, MA, USA, PN 520304).

Circulating cell-free DNA (cfDNA) was extracted from 81 plasma samples using the QIAamp Circulating Nucleic Acid Kit (Qiagen, Hilden, Germany, Cat. No. 55114) following the manufacturer’s instructions. The cfDNA was eluted in 50 µL of buffer AVE per plasma sample and concentrations were measured with the Qubit 1× dsDNA High Sensitivity Assay Kit (Thermo Fisher Scientific, Waltham, MA, USA, Cat. No. Q33265). A range of 21 ng to >2 μg cfDNA was recovered and used as input to the dPCR assay (median 267 ng).

As a germline genetic material source, DNA was extracted from whole blood for each patient using the QIAamp DNA Blood Mini Kit (Qiagen, Hilden, Germany, Cat. No. 51106).

### 2.2. Library Preparation and Sequencing

An amount of 500 ng of genomic DNA from tumor tissue was used as input for mechanical fragmentation using the Covaris E220 or ML230 instrument. WGS libraries were prepared with standard adapters for Illumina instruments. Libraries were equimolarly pooled and sequenced on the NextSeq 550 or NextSeq 500 instrument.

### 2.3. Bioinformatic Analysis

Raw sequencing reads were cleaned, quality-checked, and then aligned to the GRCh38 version of the human reference genome assembly using BWA-MEM2 [44]. Duplicates were marked following the Picard MarkDuplicates approach. Sequencing statistics were generated using Samtools and the Picard suite. SVs, SNVs, and indels were called using a proprietary workflow.

### 2.4. dPCR Assay Development and ctDNA Monitoring

Design, validation, and use of personalized dPCR fingerprints from fresh frozen tissue specimens.

Hydrolysis probe-based dPCR assays targeting up to six SVs and SAGAsafe^®^ [45,46,47,48,49,50,51], dPCR [52] assays targeting one SNV or indel per patient were designed and validated for 27 patients.

SVs were selected for inclusion based on the number of supporting reads and the probability that primers could be designed to allow the breakpoint junction to be validated by dPCR. SNVs were included if they were known hotspot mutations or occurred in common onco- or tumor suppressor genes. Orthogonal validation was performed using the patient’s tumor tissue as a positive control and matched whole blood material in order to exclude germline mutations. A maximum of four somatic SV assays and one somatic SNV were used to monitor ctDNA in the cfDNA samples extracted from plasma.

Of the 15 patients with a fresh frozen tumor tissue sequenced, the validated personalized tumor fingerprints comprised four SVs and one SNV in eleven patients, two SVs and one SNV in one patient, and four SVs alone in three patients. Of the twelve SNVs and indels included in the fingerprints, eleven were in *TP53* and one was in *ERBB2*.

For ctDNA monitoring, all cfDNA extracted was split into several dPCR reactions, each of which targets one SV or SNV within the validated tumor fingerprint of the patient (8% of the elution volume per somatic biomarker). All somatic SVs were also tracked in a multiplex dPCR reaction using 8% of the elution volume. All dPCR analyses were carried out using the QX200 droplet digital PCR (ddPCR) system (Bio-Rad, Hercules, CA, USA).

### 2.5. Design, Validation, and Use of Personalized dPCR Fingerprints from FFPE Material

Assays to detect and quantify SVs were designed and validated using SAGA Diagnostics’ proprietary technology. cfDNA analysis consisted of SV pre-amplification followed by multi-target SV detection via dPCR. Of the twelve patients from which FFPE specimens were collected and sequenced, eight patients had eight SVs passing quality control included in the dPCR fingerprint, two patients had six SVs, one patient had four SVs, and another patient had one SV tracked. cfDNA analysis was performed using QIAcuity 26k 24-well Nanoplates (Qiagen, Hilden, Germany) and QIAcuity Digital PCR Systems (Qiagen, Hilden, Germany) using tumor FFPE DNA as the positive control and sheared wildtype genomic DNA as the negative control.

### 2.6. Statistical Analysis

Statistical analysis was performed using SPSS 24.0 for Windows (IBM Corp., Armonk, NY, USA) and Microsoft Excel. Patient data were analyzed by descriptive statistics (mean and standard deviation or median and interquartile range according to distribution of values). Due to the small sample size mainly descriptive analyses and graphs are presented. Due to the skewed distribution non-parametrical tests were used for statistical analyses. To identify associations between ctDNA levels and postoperative outcome, the Wilcoxon signed rank test (paired non-parametric t-test) was performed in patients with (*n* = 8) and without (*n* = 19) postoperative RD, to see whether ctDNA levels change in preoperative and d10 samples within these two groups of patients. In addition, the Wilcoxon rank sum test was used to see whether ctDNA levels (cps/mL) in d10 samples with and without postoperative residual disease are different (*n* = 27). *p*-values of 0.05 or lower were considered statistically significant.

## 3. Results

1.Patient characteristics

Patient characteristics are provided in Table 1 and the process of recruitment in Figure 1. Seven patients (25.9%) presented with FIGO stage < 3C and 20 patients (74.1%) presented with FIGO stage ≥ 3C. Based on the surgeon’s evaluation, 19/27 patients (70.4%) had no macroscopic tumor residuals after surgery, two patients (7.5%) had RD < 10 mm, and six (22.2%) patients had RD > 10 mm, respectively.

2.Feasibility of low-coverage WGS to tailor personalized tumor-informed ctDNA assays in EOC patients

One of our main objectives was to detect ctDNA based on tumor information. We were able to assess individual SV profiles by low-coverage WGS in tumor tissue from FFPE and FF for every patient showing the universality of the approach to design somatic alteration dPCR assays using both FFPE and FF tissue. Based on this, we were able to track somatic alterations in the plasma of all patients and subsequently identify ctDNA in 96% (26/27) of patients preoperatively and in 81% (22/27) of patients at d10 (Appendix A).

3.Association of personalized tumor-informed ctDNA levels and surgeon’s evaluation of postoperative MRD

Pre- and postoperative variant allele frequencies (VAFs) are presented in Figure 2, with detection ranging from 0.000099% to 51% VAF. The relative change in ctDNA levels from preoperative to postoperative time points is shown in Figure 3 and Figure 4. Interestingly, ctDNA samples at d2 appear to be generally elevated—very likely due to tumor manipulation during surgery and subsequent release of tumor-derived DNA material into the bloodstream. Thus, all subsequent perioperative analyses compared preoperative with postoperative ctDNA samples at d10. Median postoperative ctDNA levels increased in patients with postoperative RD compared to preoperative samples in seven out of eight patients (*p* = 0.01563). One patient with postoperative RD had decreased ctDNA levels at d10. This patient had postoperative RD of <5 mm at the end of surgery. Median postoperative ctDNA levels decreased significantly in 17/19 patients without postoperative RD (*p* = 0.0007983). Thus, there was a significant difference for median ctDNA levels at d10 between patients with and without postoperative RD (367.38 cps/mL (2.84% VAF) and 0.92 cps/mL (0.017% VAF), respectively). Patients with postoperative RD had significantly higher ctDNA levels (*p* = 0.0003972) (Figure 3).

The evaluation of postoperative ctDNA levels in patients stratified by tumor stage (FIGO stage < 3C vs. FIGO stage ≥ 3C) is of particular interest, illustrated in Figure 4. In patients with surgeon-reported complete tumor resection (*n* = 19), 71% (five out of seven) of patients with FIGO stage < 3C compared to 42% (five out of 12) of patients with FIGO stage ≥ 3C presented with a decrease in postoperative ctDNA levels. Interestingly, the decline in ctDNA levels seemed to be more rapid and pronounced in patients with FIGO stage < 3C than in patients with FIGO ≥ 3C (Figure 4).

## 4. Discussion

In this prospective study, the present tumor-informed ctDNA approach utilizing personalized SVs and SNVs seems feasible providing high pre- and postoperative detection rates. The applied technique is highly sensitive, as ctDNA was detected in 96.3% (26/27) patients before surgery at detection limits as low as 0.00009% VAF. Postoperative ctDNA levels at d10 after surgery differed substantially based on postoperative RD: in patients with postoperative RD median postoperative ctDNA levels increased compared to preoperative measurements; in patients without postoperative RD, median postoperative ctDNA levels decreased significantly compared to preoperative measurements. Moreover, in patients with very advanced disease, i.e., FIGO stage ≥ 3C and surgeon-reported complete tumor resection, ctDNA levels could potentially serve as a triage test to identify patients with complete resection and patients with MRD.

Standard-of-care postoperative surveillance is limited to imaging- and/or blood-based biomarkers such as CA-125, that are a proxy for ongoing disease but have demonstrated poor reproducibility for assessing MRD [53]. Growing evidence demonstrates that detection of MRD by ctDNA seems to identify a group of patients who are at high risk for recurrence in several cancer types [54]. Literature on this topic is growing for various tumors, solid as well as non-solid [15,16,55,56,57,58,59,60,61,62,63]. Within our cohort, data suggest that the present tumor-informed ctDNA approach could be sensitive enough to detect MRD. This is supported by the fact that in earlier advanced EOC, i.e., FIGO stage < 3C, the concordance between complete resection and decrease in ctDNA levels was extremely high. In very-advanced EOC, i.e., FIGO stage ≥ 3C, with surgeon-reported complete resection, there were two groups of patients: one group of patients with significantly decreased postoperative ctDNA levels and another group of patients with only slightly decreased postoperative ctDNA levels. This might be a first hint that ctDNA levels could be more accurate to detect MRD than the surgeon’s assessment at the end of surgery. This could allow the identification of a subset of patients at high risk for recurrence despite having had surgeon-reported complete resection. This could ultimately identify patients in need for more extensive maintenance therapy after chemotherapy [53].

Previous studies investigated ctDNA levels perioperatively in EOC [39,61,64,65]. Preoperative detection rates ranged from 60% to 100% [39,61,64,65,66]. Three studies used a tumor-informed ctDNA approach with preoperative detection rates ranging from 60% to 100% [39,64,65]. Thus, our study—also using a tumor-informed ctDNA approach—is supporting this approach with a very high preoperative detection rate of >95%. The present ctDNA approach used low-coverage WGS on tumor tissue to identify individual rearrangement fingerprints to tailor up to eight personalized ddPCR-SV assays per patient. Moreover, the present study applied very stringent criteria for patient inclusion to provide a very homogenous patient cohort.

With respect to perioperative ctDNA evaluation only one other study has investigated tumor-informed ctDNA levels in advanced EOC in a comparable setting [64]. This study comprised 20 patients, in which FFPE tumor samples and ctDNA samples were compared to detect key features, of use to identify ctDNA. Within the study by Heitz et al., only nine out of 20 (45%) patients had detectable ctDNA levels postoperatively with 50% of patients having had complete macroscopic tumor resection during surgery. This is in contrast with our study, in which 81% of patients had detectable ctDNA levels postoperatively with 70% of patients having had complete macroscopic tumor resection during surgery. This difference in detection rates seems to be caused by the different methods applied for ctDNA detection. In our study, an extremely sensitive detection method with detection limits as low as 0.00009% VAF and the ability to load high inputs of cfDNA to dPCR (up to 2 μg herein) was applied. This enables the detection of very-low mutant VAF and the impact of high levels of normal background post-surgery can be overcome. Another study evaluated postoperative tumor-informed ctDNA during chemotherapy for treatment response [67]. The postoperative detection rate before chemotherapy was 82.9% (39/47 patients)—information on preoperative ctDNA detection rates is lacking [67]. These rates are in line with our findings and suggest that the technology applied in the present manuscript seems to be one of the most promising to identify MRD.

Strengths of this study comprise the use of prospective samples assessed from all patients at the exact same perioperative time points (before surgery, day 2 and day 10 after surgery), a very homogenous cohort of patients (only advanced tumor stage III–IV, primary surgery without the use of neoadjuvant chemotherapy, and restriction to high-grade serous ovarian cancer histology), and strict evaluation of postoperative tumor residuals due to the prospective nature of the study. Limitations of the study comprise the limited sample size of the present cohort, which currently does not allow the determination of solid postoperative ctDNA cut-off levels or algorithms to reliably confirm complete resection after surgery. Moreover, long-term follow-up for the cohort is currently pending, which does not allow the evaluation of the prognostic implication of postoperative ctDNA plasma levels. Of note, a larger prospective multicenter study is currently underway to address these questions.

## 5. Conclusions

The surgeon’s evaluation is currently the standard of care to assess the amount of postoperative residual disease. Our findings suggest that the surgeon’s assessment of “no MRD” might be more precise in patients with FIGO stage < 3C as there is smaller risk to miss tumor residuals. In contrast, the surgeon’s evaluation in patients with FIGO stage ≥ 3C might be more imprecise. The present tumor-informed ctDNA approach seems to be feasible and highly sensitive. Since a certain ctDNA level was also detectable in patients with tumor stage FIGO ≥ 3C who were classified as “completely resected”, our study provides first evidence that ctDNA might be sensitive enough to detect MRD, which might not be visible to the surgeon’s eye—either because of size or because of localization. Detection of MRD could lead to an improved triage for (de)escalation of postoperative systemic therapy. Consequently, our method for ctDNA detection and quantification holds immense potential for early detection of residual disease, accurate diagnosis—in particular, in the evaluation of the most important prognostic factor, the postoperative residual disease—treatment monitoring, and personalized treatment strategies. Our results are in line with promising approaches and findings of groups investigating other malignancies [64,67,68]. In parallel, a prospective multicenter study is underway to address the limitations of the present feasibility pilot study.

## Figures and Tables

**Figure 1 cancers-17-00786-f001:**
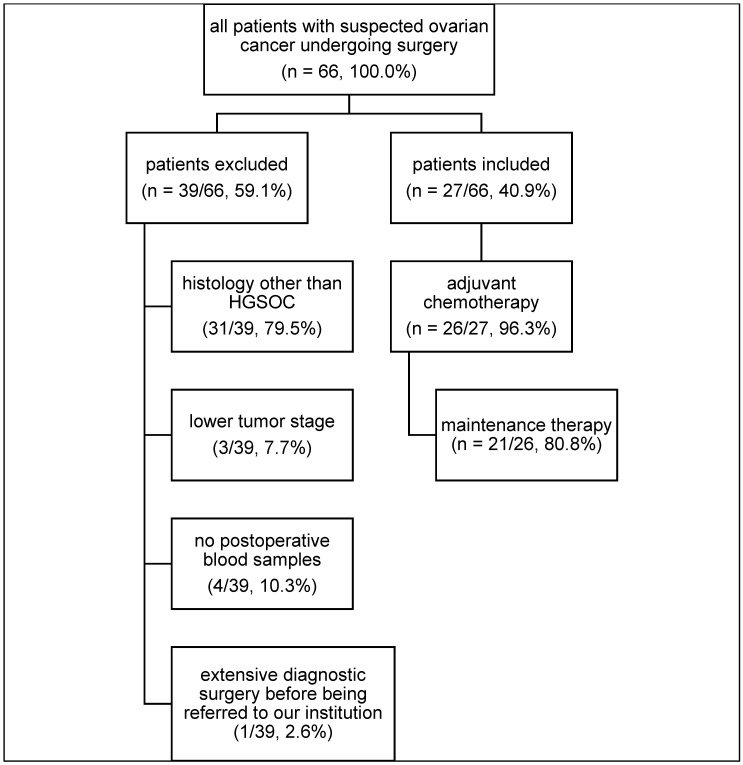
CONSORT diagram including all patients with suspicion for newly diagnosed advanced high-grade serous ovarian cancer undergoing primary surgery.

**Figure 2 cancers-17-00786-f002:**
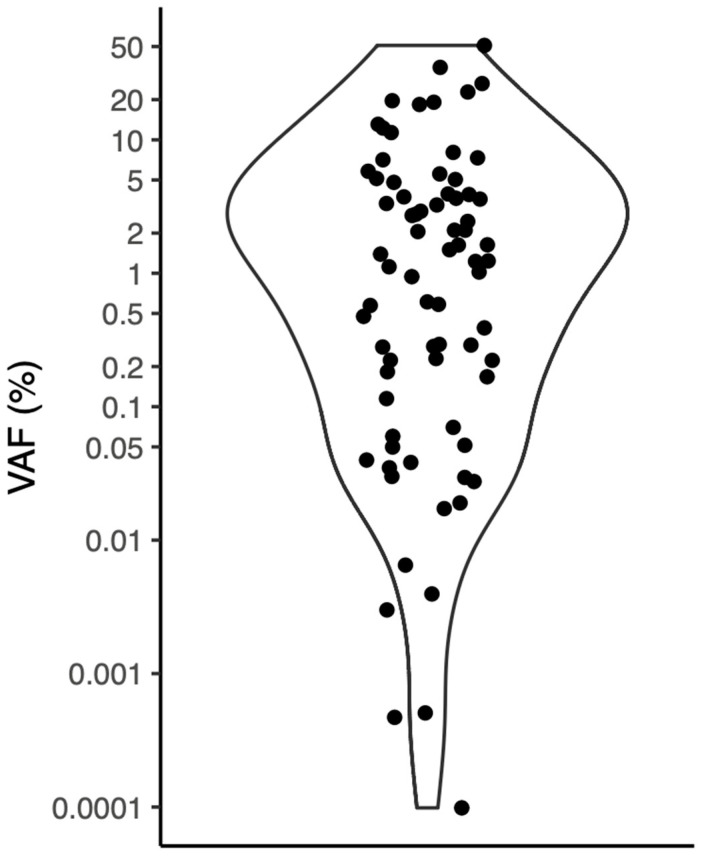
Range of ctDNA detection levels (% variant allele frequency; VAF) in all plasma samples collected at pre- and postoperative time points.

**Figure 3 cancers-17-00786-f003:**
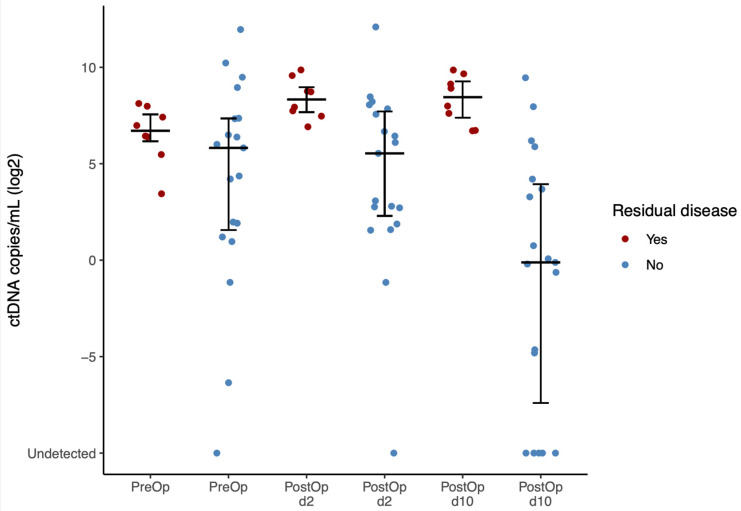
Circulating tumor DNA (ctDNA) levels (copies/mL) in primary advanced high-grade serous ovarian cancer patients preoperatively (PreOp) and postoperatively (PostOp; postoperative day 2, d2, and 10, d10) broken down by surgeon-reported postoperative residual disease at end of surgery.

**Figure 4 cancers-17-00786-f004:**
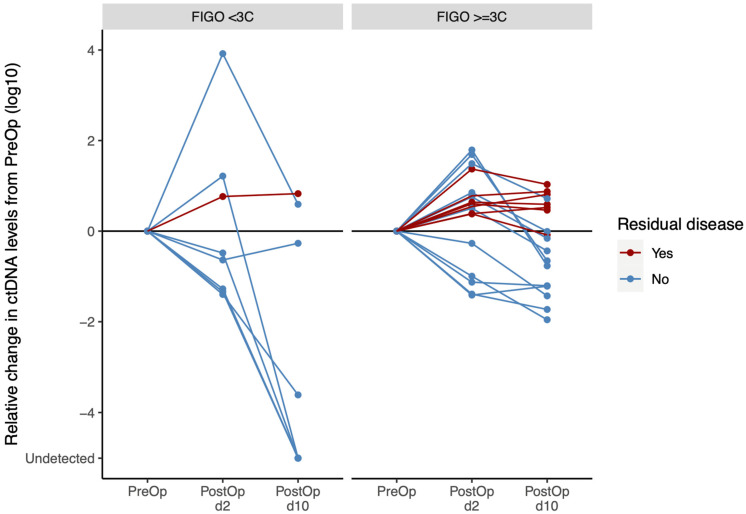
Relative change in ctDNA levels (copies/mL) from preoperative (PreOp) levels to postoperative day 2 and 10 (PostOp; d2, d10) levels in primary advanced high-grade serous ovarian cancer patients broken down by postoperative residual disease evaluated by the surgeon at the end of surgery and tumor stage (FIGO stage < 3C vs. FIGO stage ≥ 3C).

**Table 1 cancers-17-00786-t001:** Patient characteristics for patients undergoing surgery during their treatment for primary advanced high-grade serous ovarian cancer.

Patients		N = 27	100%
age (years)	mean 64 (39–80)		
FIGO tumor stage	IIIA1	2	7.4%
	IIIB	5	18.5%
	IIIC	15	55.6%
	IVA	1	3.7%
	IVB	4	14.8%
HRD status	negative	13	48.1%
	positive	12	44.4%
	n.a.	2	7.4%
sBRCA status	negative	20	74.1%
	positive	5	18.5%
	n.a.	2	7.4%
postoperative tumor residuals	no	19	70.4%
	1–10 mm	2	7.4%
	>10 mm	6	22.2%

HRD = homologous recombination deficiences; BRCA = somatic BRCA; n.a. = not applicable.

## Data Availability

The data presented in this study are available on request from the corresponding authors.

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
