# Peer review of "ctDNA as an Objective Marker for Postoperative Residual Disease in Primary Advanced High-Grade Serous Ovarian Cancer"

_cancers, 2025, doi:10.3390/cancers17050786_

Round 1
Reviewer 1 Report
Comments and Suggestions for Authors
The aim of the study presented in the manuscript was to investigate the feasibility of circulating tumor DNA (ctDNA) as an objective marker for postoperative RD. In this prospective study, 27 patients with advanced EOC undergoing primary surgery between July 2021 and July 2022 were included. The authors analyzed Blood samples at day 2 (d2) and 10 (d10). The authors performed Low-coverage whole genome sequencing (WGS) to identify structural variants (SVs) at single-base pair resolution, and they have made attempts to use single nucleotide variants (SNVs) and indels in tumor tissue to develop personalized, tumor-informed digital polymerase chain reaction (dPCR) fingerprint assays for each patient. The authors report that they were able to develop the dPCR fingerprint assays successfully developed for all patients by identifying one to eight SVs/SNVs per patient. They are also reporting that the ctDNA was detected in 96% (n=26/27) of patients preoperatively and in 81% (n=22/27) of patients at d10. Median ctDNA levels at d10 were significantly higher in patients with postoperative RD (median 367.38 copies (cps)/mL, 2.84% variant allele frequency; VAF) than in patients without postoperative RD (median 0.92 cps/mL, 0.017% VAF, p < 0.001). The authors have found that in patients with postoperative RD, ctDNA levels increased from preoperative to d10 in 7/8 patients (p = 0.016). In patients with complete tumor resection, ctDNA levels decreased from preoperative to d10 in 17/19 patients (p < 0.001).The authors are confirming that personalized ctDNA approach demonstrated feasibility, providing extremely high detection rates pre- and postoperatively. They are also suggesting that their results indicate that this approach could potentially be used for postoperative RD assessment in patients with primary advanced EOC.
A major concern of this manuscript is that the study patients were recruited in 2021-2022 in this prospective study. Since the recruitment was completed in 2022 it makes us to wonder shy short-term and long- term follow-up was done. It creates a curiosity to know what happens to the ctDNA after d10 of the postoperative period. Interestingly, the authors themselves alluding to the fact that “the long-term follow-up for the cohort is currently pending, which does not allow the evaluation of the prognostic implication of postoperative ctDNA”.
In addition, the advantages of the detection method lies in the methodology utilized because the authors are stating that “tumor-informed ctDNA detection approach has proven to be feasible and highly sensitive”. They are also claiming that, a certain ctDNA level was also detectable in patients with tumor stage FIGO >3C who were classified as “completely resected”, their study provides the first evidence that ctDNA might be sensitive enough to detect MRD. Since the methodology is not detailed in this manuscript and also because the manuscript emphasizing the details of the methodology is under preparation, this manuscript, if published, can be only informative.
The authors have gone through the effort of completing the SV, SNV and Indel analyses. It would be more supportive to the manuscript, if they can provide the table of the significant SV, SNV or Indel that is correlated to the levels of postoperative ctDNA.
Overall, this manuscript contains clinical important information and emphasizes the benefits of tumor-informed ctDNA detection approach and its advantages compared to the postoperative evaluation of the MRD by the surgeon. However, this requires additional improvement with the language and the data included in this manuscript. Some of the minor comments for consideration are included below.
Minor comments:
L18: The word ‘evaluate is used repeatedly in the sentence. Modify one of them.
L20: This sentence is clear. Rephrasing is highly recommended and introduce the abbreviation ctDNA
L29: Remove the wording ‘circulating tumor DNA’ keep the abbreviation
L59: The abbreviation for Residual Disease is already established. Use RD here to minimize confusion.
L61: Rephrasing the sentence as “……..all currently available clinical tests,……….’ Can be considered
L77: Authors must specify what imaging
L102: (preop) can be removed.
L127: Modify as ‘DNA / RNA’ and indicated by the manufacturer
L157: Have they submitted George et al, manuscript?
L262: The resolution of CONSOR diagram is poor and font size is msall. Needs improvement
L339: This part of the paragraph has redundancy and complex long sentence. Authors need to simplify the sentences.
L350 – L353: While it is appropriate to highlight the advantages of the ctDNA detection method discussed in this manuscript, the sentences in this section sounds more critical of Heitz et al method. Instead, the difference in the technologies can be elaborated with a positive light.
L360: Information provided in this paragraph can me moved to the beginning of the discussion
L394: “Might be more flawed” can be rephrased as “might be more imprecise”
Comments on the Quality of English LanguageThe manuscript uses long sentences with redundancy. It reads more like a case study than a scientific manuscript. Rephrasing the sentences and thorough proof reading may make this manuscript better
Author Response
Comments 1:
A major concern of this manuscript is that the study patients were recruited in 2021-2022 in this prospective study. Since the recruitment was completed in 2022 it makes us to wonder shy short-term and long- term follow-up was done. It creates a curiosity to know what happens to the ctDNA after d10 of the postoperative period. Interestingly, the authors themselves alluding to the fact that “the long-term follow-up for the cohort is currently pending, which does not allow the evaluation of the prognostic implication of postoperative ctDNA”.
Response 1:
The reviewer is mentioning a very important point, as the prognostic implication is a crucial aspect. Follow-up of patients and long-term follow-up analyses of ctDNA levels have currently not been incorporated in the present manuscript for following reasons: 1. The scope of this pilot cohort was to demonstrate feasibility by measuring ctDNA around the time of maximal change of tumor load, i.e. before and after debulking surgery; 2. due to heterogenous treatment modalities – particularly with respect to maintenance therapy – the respective cohorts during follow-up are extremely small and heterogenous and therefore nearly impossible to analyze or draw relevant conclusions; 3. as the results of the present feasibility cohort are promising a larger validation cohort is currently underway to confirm these first promising results and address the prognostic implication.
Comments 2:
In addition, the advantages of the detection method lie in the methodology utilized because the authors are stating that “tumor-informed ctDNA detection approach has proven to be feasible and highly sensitive”. They are also claiming that, a certain ctDNA level was also detectable in patients with tumor stage FIGO >3C who were classified as “completely resected”, their study provides the first evidence that ctDNA might be sensitive enough to detect MRD. Since the methodology is not detailed in this manuscript and also because the manuscript emphasizing the details of the methodology is under preparation, this manuscript, if published, can be only informative.
Response 2:
We appreciate the comments from the reviewers and the opportunity to clarify the methods. We have removed the reference to George et al, in preparation, and added reference to the now published manuscript detailing SAGAsafe assays (Hill et al, Nature 616, 159–167 (2023)) at line 177.
We have also expanded the section relating to methods from FFPE and added the following:
“Assays to detect and quantify SVs were designed and validated using SAGA Diagnostics’ proprietary technology. cfDNA analysis consisted of SV pre-amplification followed by multi-target SV detection via dPCR. Of the twelve patients from which FFPE specimens were collected and sequenced, eight patients had eight SVs passing quality control included in the dPCR fingerprint, two patients had six SVs, one patient had four SVs and another patient had one SV tracked. cfDNA analysis was performed using QIAcuity 26k 24-well Nanoplates (Qiagen) and QIAcuity Digital PCR Systems (Qiagen) using tumor FFPE DNA as positive control and sheared wildtype genomic DNA as negative control." (see line 200 ff.)
Comments 3:
The authors have gone through the effort of completing the SV, SNV and Indel analyses. It would be more supportive to the manuscript, if they can provide the table of the significant SV, SNV or Indel that is correlated to the levels of postoperative ctDNA.
Response 3:
To provide additional information on the SVs and SNVs detected, we have included an additional supplementary table showing the co-ordinates of each SV or SNV targeted and the detection at each plasma time point. We have included a reference to this table at line 256: “We were able to assess individual SV profiles by low-coverage WGS in tumor tissue from FFPE and FF for every patient showing the universality of the approach to design somatic alteration dPCR assays using both FFPE and FF tissue. Based on this, we were able to track somatic alterations in the plasma of all patients and subsequently identify ctDNA in 96% (26/27) of patients preoperatively and in 81% (22/27) of patients at d10 (Supplementary tables 1 and 2“ (see line 250 ff.)
Comments 4:
L18: The word ‘evaluate is used repeatedly in the sentence. Modify one of them.
Response 4:
Thank you for pointing this out. We have changed the relevant section (see line 18).
Comments 5:
L20: This sentence is clear. Rephrasing is highly recommended and introduce the abbreviation ctDNA
Response 5:
Thank you for your comment. We agree and corrected it accordingly (see line 21).
Comments 6:
L29: Remove the wording ‘circulating tumor DNA’ keep the abbreviation
Response 6:
Thank you, we corrected this (see line 29).
Comments 7:
L59: The abbreviation for Residual Disease is already established. Use RD here to minimize confusion.
Response 7:
We agree and adapted accordingly (see line 70).
Comments 8:
L61: Rephrasing the sentence as “……..all currently available clinical tests,……….’ Can be considered
Response 8:
We rephrased the sentence, see line 72 ff.
Comments 9:
L77: Authors must specify what imaging
Response 9:
Thank you for your recommendation. We have provided the information that the data relates to CT scans (see line 88).
Comments 10:
L102: (preop) can be removed.
Response 10:
Thank you, we have removed it, see line 120.
Comments 11:
L127: Modify as ‘DNA / RNA’ and indicated by the manufacturer
Response 11:
The ctDNA assay used in this study only required DNA from FFPE or FF material, therefore the DNA only protocol was followed according to the manufacturer's instructions. We have added “for DNA only extraction” to line 145 ff.
Comments 12:
L157: Have they submitted George et al, manuscript?
Response 12:
We have removed the reference to George et al, in preparation, and added reference to the now published manuscript detailing SAGAsafe assays (Hill et al, Nature 616, 159–167 (2023)) at line 177.
Comments 13:
L262: The resolution of CONSOR diagram is poor and font size is small. Needs improvement
Response 13:
We agree and have improved the figure accordingly (see line 245)
Comments 14:
L339: This part of the paragraph has redundancy and complex long sentence. Authors need to simplify the sentences.
Response 14:
Thank you for your comment. We have amended the corresponding passage (see line 494 ff.)
Comments 15:
L350 – L353: While it is appropriate to highlight the advantages of the ctDNA detection method discussed in this manuscript, the sentences in this section sounds more critical of Heitz et al method. Instead, the difference in the technologies can be elaborated with a positive light.
Response 15:
We acknowledge this comment and therefore rephrased the whole paragraph to be more precise on the applied technology and discuss the results of the most recent publications (see line 475 ff.).
Comments 16:
L360: Information provided in this paragraph can me moved to the beginning of the discussion
Response 16:
We agree and have revised the discussion (see line 364 ff.).
Comments 17:
L394: “Might be more flawed” can be rephrased as “might be more imprecise”
Response 17:
Thank you for pointing this out. We rephrased the sentence, see line 610.
Reviewer 2 Report
Comments and Suggestions for Authors
The manuscript “ctDNA as an objective marker for postoperative residual disease in primary advanced high-grade serous Ovarian Cancer”, although interesting, requires major revisions to address both content-related issues and compliance with the journal's guidelines.
1. Minor English editing is needed. The use of abbreviations throughout the manuscript is inconsistent. For example, the authors did not define the abbreviation ctDNA in the summary section. Also, they mention VAF (variant allele frequency) in the abstract section but did not define it first. Abbreviations should be defined at first use and used consistently throughout the manuscript.
2. Abstract: The authors should indicate the range of tumour stages in the abstract section, particularly considering the small sample size.
3. Introduction: “Ovarian cancer (OC) is the leading cause of mortality in gynecological malignancies”. This statement is incorrect. OC is the most lethal gynaecological tumour but not the leading cause of mortality due to gynaecological malignancy in most countries. The authors should clarify the sentence.
4. Introduction: “Thus, we investigated the feasibility and accuracy of a tumor-informed personalized ctDNA approach utilizing monitoring of structural rearrangements and single nucleotide variants (SNVs) in advanced EOC patients for the measurement of postoperative RD, comparing it with the current gold standard, i.e. surgeon´s evaluation at the end of debulking surgery”. The authors should change the sentence for “…. monitoring of somatic structural rearrangements…”.
5. Methodology: The authors used two DNA extraction methods considering the available samples. The authors should discuss the implications.
6. Methodology: “A range of 21 ng to >2mg cfDNA was recovered and used as input to the dPCR assay (median 267 ng)”. The authors did not discuss normalisation measures for the dPCR analyses.
7. Methodology: In line 170, the gene names should be formatted in italics to adhere to standard scientific conventions.
8. Methodology: the statistical analysis section is missing some important information. For instance, what was the test used to assess data normality?
9. Results: “Seven patients (25.9%) presented with FIGO stage <3C and 20 patients (74.1%) presented with FIGO stage >3C.” Considering the phrase, I assumed that no patient had FIGO stage 3C, which is incorrect (FIGO stage ≥3C). The authors should clarify this point throughout the manuscript.
10. Results: The authors mentioned adjuvant chemotherapy and maintenance therapy in the population characterisation but did not use that information in data analysis. Perhaps the data could be more important for the larger prospective multicenter study currently underway referenced in the discussion section.
11. Figures and tables: The authors should follow the journal guidelines. Figures and tables should be presented close to the relevant text where they are first mentioned. Figure 1 requires editing in terms of colour and letter size. Also, the name of the figures should appear below the figures not above. Figure 2 was mentioned two times in the same sentence (line 216).
12. Conclusions: “The present tumor-informed ctDNA approach has proven to be feasible and highly sensitive.” Considering the small cohort size, this statement should be carefully reconsidered. A more cautious phrasing would better reflect the study's scope and limitations.
13. Declarations sections: I noticed the absence of key sections, including the Data Availability Statement and Acknowledgments. I also suggest the authors add the “Abbreviations” section.
14. References: Considering the evolution of OC treatment, the authors should cite more up-to-date references (for instance: https://doi.org/10.3390/ijms25031845; https://doi.org/10.1002/ctm2.70012). Recently a paper on ctDNA levels in OC patients was published (https://doi.org/10.1016/j.ygyno.2024.11.002). I suggest the authors discuss this work in the discussion section.
Comments on the Quality of English LanguageThe manuscript requires minor English editing.
Author Response
Comments 1:
Minor English editing is needed. The use of abbreviations throughout the manuscript is inconsistent. For example, the authors did not define the abbreviation ctDNA in the summary section. Also, they mention VAF (variant allele frequency) in the abstract section but did not define it first. Abbreviations should be defined at first use and used consistently throughout the manuscript.
Response 1:
Thank you for the comment. We corrected this, see line 21 and line 39.
Comments 2:
Abstract: The authors should indicate the range of tumour stages in the abstract section, particularly considering the small sample size.
Response 2:
We agree on this comment and have therefore added the information accordingly (see line 30).
Comments 3:
Introduction: “Ovarian cancer (OC) is the leading cause of mortality in gynecological malignancies”. This statement is incorrect. OC is the most lethal gynaecological tumour but not the leading cause of mortality due to gynaecological malignancy in most countries. The authors should clarify the sentence.
Response 3:
We thank the reviewer for correcting this statement. We have clarified this section accordingly, see line 62).
Comments 4:
Introduction: “Thus, we investigated the feasibility and accuracy of a tumor-informed personalized ctDNA approach utilizing monitoring of structural rearrangements and single nucleotide variants (SNVs) in advanced EOC patients for the measurement of postoperative RD, comparing it with the current gold standard, i.e. surgeon´s evaluation at the end of debulking surgery”. The authors should change the sentence for “…. monitoring of somatic structural rearrangements…”.
Response 4:
Thank you for pointing this out. We adapted it accordingly (see line 98).
Comments 5:
Methodology: The authors used two DNA extraction methods considering the available samples. The authors should discuss the implications.
Response 5:
We agree that this is an important point and useful to highlight that both FF and FFPE material are suitable starting materials and that either is feasible for generating a tumour informed fingerprint. These tissue types require different extraction chemistries. We have updated the language in the results to highlight that successful panels could be designed for FFPE and FF. “Feasibility of low-coverage WGS to tailor personalized tumor-informed ctDNA assays in EOC patients. One of our main objectives was to detect ctDNA based on tumor information. We were able to assess individual SV profiles by low-coverage WGS in tumor tissue from FFPE and FF for every patient. Showing the universality of the approach to design somatic alteration dPCR assays using both FFPE and FF tissue. Based on this, we were able to track somatic alterations in the plasma of all patients and subsequently identify ctDNA in 96% (26/27) of patients preoperatively and in 81% (22/27) of patients at d10.“ (See line 250 ff.)
Comments 6:
Methodology: “A range of 21 ng to >2mg cfDNA was recovered and used as input to the dPCR assay (median 267 ng)”. The authors did not discuss normalisation measures for the dPCR analyses.
Response 6:
The inputs into dPCR, in terms of DNA mass amounts, were not normalized and the entire sample amount added into the assay. The dPCR results were reported as copies/mL of plasma and variant allele frequencies. These values were calculated using the raw dPCR measurements and are given in Supplementary table 1. We have clarified at line 193 of the methods that all cfDNA was loaded into the assay “For ctDNA monitoring, all cfDNA extracted was split into several dPCR reactions“
Comments 7:
Methodology: In line 170, the gene names should be formatted in italics to adhere to standard scientific conventions.
Response 7:
Thank you for the comment, we corrected it (see line 191).
Comments 8:
Methodology: the statistical analysis section is missing some important information. For instance, what was the test used to assess data normality?
Response 8:
We thank the reviewer for pointing out the statistical limitations. Of note, we therefore used mainly descriptive analyses and graphs to acknowledge the limitations of a feasibility study with a small number of patients. In the limited situations, in which statistical tests were applied, we used non-parametrical tests (the Wilcoxon tests) due to the small sample size and the skewed distribution. This has now been added to the “statistical analysis” section (see line 219 ff.)
Comments 9:
Results: “Seven patients (25.9%) presented with FIGO stage <3C and 20 patients (74.1%) presented with FIGO stage >3C.” Considering the phrase, I assumed that no patient had FIGO stage 3C, which is incorrect (FIGO stage≥3C). The authors should clarify this point throughout the manuscript.
Response 9:
Thank you for this important information. We have corrected it throughout the manuscript.
Comments 10:
Results: The authors mentioned adjuvant chemotherapy and maintenance therapy in the population characterisation but did not use that information in data analysis. Perhaps the data could be more important for the larger prospective multicenter study currently underway referenced in the discussion section.
Response 10:
The reviewer is right in pointing out, that this information is not really necessary for describing the setting within the present feasibility study. Thus, we deleted this information from Table 1 (see line 240).
Comments 11:
Figures and tables: The authors should follow the journal guidelines. Figures and tables should be presented close to the relevant text where they are first mentioned. Figure 1 requires editing in terms of colour and letter size. Also, the name of the figures should appear below the figures not above. Figure 2 was mentioned two times in the same sentence (line 216).
Response 11:
All these comments are very valuable and have been taken into account. Thank you very much.
Comments 12:
Conclusions: “The present tumor-informed ctDNA approach has proven to be feasible and highly sensitive.” Considering the small cohort size, this statement should be carefully reconsidered. A more cautious phrasing would better reflect the study's scope and limitations.
Response 12:
We agree with the reviewer’s comment and therefore rephrased this sentence acknowledging the feasibility setting of our study (see line 610).
Comments 13:
Declarations sections: I noticed the absence of key sections, including the Data Availability Statement and Acknowledgments. I also suggest the authors add the “Abbreviations” section.
Response 13:
Thank you for your comment. We added the mentioned sections (see line 666 ff.).
Comments 14:
References: Considering the evolution of OC treatment, the authors should cite more up-to-date references (for instance: https://doi.org/10.3390/ijms25031845; https://doi.org/10.1002/ctm2.70012). Recently a paper on ctDNA levels in OC patients was published (https://doi.org/10.1016/j.ygyno.2024.11.002). I suggest the authors discuss this work in the discussion section.
Response 14:
Thank you so much for this important input. We added the suggested references.
Round 2
Reviewer 2 Report
Comments and Suggestions for Authors
I want to thank the authors for revising their manuscript and improving its quality. Beyond English editing, I have only a comment to address. As previously mentioned, the figures and tables should be presented close to the main text where they were first mentioned. Also, while the name/caption of the figures should appear below the figures, the tables' names must be placed above the tables.
Comments on the Quality of English LanguageThe manuscript requires English editing.